# Quenched Flux-Coupling Superconducting Fault Current Limiter Scheme and Its Electromagnetic Design Method

**DOI:** 10.3390/ma16020754

**Published:** 2023-01-12

**Authors:** Sinian Yan, Li Ren, Jinghong Zhao, Ying Xu, Shifeng Shen, Yiyong Xiong, Baolong Liu, Feiran Xiao

**Affiliations:** 1School of Electrical Engineering, Naval University of Engineering, Wuhan 430033, China; 2State Key Laboratory of Advanced Electromagnetic Engineering and Technology, Huazhong University of Science and Technology, Wuhan 430074, China

**Keywords:** electromagnetic design, parallel inductance, quenched FC-SFCL, simplified calculation method

## Abstract

In order to solve the problem of excessive short-circuit current in the present power system, a fault current limiter has become a new type of power device with high demand and is one of the current research hotspots. The flux-coupling type superconducting fault current limiter (FC-SFCL) generates a current-limiting impedance through decoupling superconducting parallel inductance based on the circuit breakers’ fractional interruption. The principle is simple, and the impedance is low during normal operation. It can directly use the existing circuit breaker to open a short circuit that is much higher than its own breaking capacity. Thus, it can be used for large-capacity fault current limiting and effective failure breaking. This paper focused on exploring and studying the implementation scheme of practical products of FC-SFCL. Considering that the quenched-type parallel inductance can limit the first peak value of the fault current, a quenched-type improvement scheme was proposed. Then, an electromagnetic design method based on the simplified calculation of the number of parallel tapes was proposed, which simplified the design process and reduced the design difficulty of the quenched FC-SFCL. Taking a 10 kV/500 A/5 kA quenched prototype as an example, its electromagnetic design was completed, and the performances of the non-quenched and quenched schemes were compared. The results showed that, compared to the non-quenched structure, the technical economics of the quenched one were more prominent, and it can be used preferentially for engineering prototypes. This study about the scheme of the quenched FC-SFCL and its electromagnetic design method is useful for promoting the implementation of the current limiter engineering prototype.

## 1. Introduction

In the modern power system with a large scale and high reliability requirements, an excessive short-circuit current may not only cause the system to lose stability due to the difficulty of removing the fault, but it can also damage the power installation due to the generated electromagnetic force and temperature rise. In order to solve the problem of the excessive short-circuit current in the present power system, a fault current limiter has become a new type of power device with high demand and is one of the current research hotspots [1]. Based on the characteristics of superconductors such as zero resistance, high current carrying, and fast state transition, the superconducting fault current limiter (SFCL) has been widely studied for its advantages of active triggering and resetting and fast current limiting speed [2,3]. Many kinds of SFCLs have been developed, such as the resistive type [4], bridge type [5], saturated iron-core type [6], flux-lock type [7], and so on.

The flux-coupling type superconducting fault current limiter (FC-SFCL) generates a current-limiting impedance through the decoupling superconducting parallel inductance based on the circuit breakers’ fractional interruption [8]. The principle is simple, and the impedance is low during normal operation. It can directly use the existing circuit breaker to open a short circuit that is much higher than its own breaking capacity. It provides a scheme to solve the technical problem of the excessive short-circuit current and circuit breaker breaking difficulty. There is a lot of research on FC-SFCL in power system application scenarios [9,10,11], parameter matching [12], operation control [13], etc. However, the magnetic-flux-reversed coupling parallel inductance and its operating loss are still to be further studied. 

This paper aimed to promote the engineering application of the FC-SFCL, focusing on superconducting parallel inductance, and studied several technical issues related to engineering application. Considering that the non-quenched FC-SFCL cannot limit the first peak value of the fault current, a quench-type improvement scheme was proposed, and its equivalent circuit and working principle were introduced. Then, the validity of the method of calculating the number of parallel tapes of the parallel inductance, ignoring superconducting characteristics and using normally conducting wires, was proven from the perspective of tapes and coils. A design method based on the simplified calculation of the number of tapes in parallel to the quenched FC-SFCL was proposed. The electromagnetic design of 10 kV/500 A/5 kA quenched FC-SFCL prototypes was completed, and the performances of the non-quenched and quenched schemes were compared.

## 2. Operating Principle and Equivalent Circuit

### 2.1. Operating Principle of the Quenched FC-SFCL

Considering that the non-quenched FC-SFCL [14] cannot limit the first peak value of the fault current, a quench-type improvement scheme was proposed that allows the parallel inductance to quench. This scheme can not only limit the first peak value of the fault current to a certain extent and improve the overall current limiting capability, but it can also greatly reduce the number of tapes used and improve the technical and economic efficiency. After the power system is short-circuited, the parallel inductance can produce a quench resistance to limit the first peak of the fault current. After the parallel inductance is decoupled, the current-limiting inductance and the quench resistance jointly limit the current. The current-limiting effect of the quench resistance of the parallel inductance further reduces the interruption requirements of the main and auxiliary circuit breakers. The current-limiting function of the quench resistance and the operation of the main and auxiliary circuit breakers at different times form the current-limiting process of the current limiter.

### 2.2. Equivalent Circuit Analysis of the Quenched FC-SFCL

After the parallel inductance of the quench-type FC-SFCL is decoupled, the equivalent circuit of the parallel inductance is shown in Figure 1a, and the equivalent schematic diagram of the access line is shown in Figure 1b. Among them, *R*_a_ and *R*_b_ represent the quench resistance of the two superconducting coils after quench, *L*_1_ and *L*_2_ are the self-inductance of the two superconducting coils, and *M* is the mutual inductance, respectively.

As for the quench resistance of the two superconducting parallel coils, there is *R*_a_ = *R*_b_ = 0 in the rated state and *R*_a_ ≠ 0, *R*_b_ ≠ 0 after the superconducting coils quench. In the rated state and the fault state, the equivalent impedance *Z*_eq_ of the parallel inductance shown as Figure 1a can be expressed as
(1)$$Z_{\mathrm{eq}}=\frac{R_{a}R_{b}+R_{a}j\omega L_{2}+R_{b}j\omega n^{2}L_{2}+\omega^{2}(k^{2}-1)n^{2}L_{2}{}^{2}}{R_{a}+R_{b}+j\omega (n^{2}+2kn+1)L_{2}}$$
where $k=M/\sqrt{L_{1}L_{2}}$ and $n=\sqrt{L_{1}/L_{2}}$ are, respectively, the coupling coefficient and the transformation ratio of the windings. If the coupling coefficient of the two parallel windings are close to 1, that is, *k*_2_ ≈ 1, then the equivalent impedance *Z*_eq_ is about 0 in the rated state, and *Z*_eq_ has a certain resistance value in the fault state.

After the parallel windings are decoupled, the current-limiting equivalent impedance *Z*_SFCL_ greatly increases, which can be expressed as
(2)$$\begin{array}{l} Z_{\mathrm{SFCL}}{}^{*}=R_{b}+\frac{\left(R_{1}+R_{a})j\omega L_{2}+{\left(n\omega L_{2}\right)}^{2}(k^{2}-1\right)}{R_{1}+R_{a}+n^{2}j\omega L_{2}}\\ Z_{\mathrm{SFCL}}=Z_{\mathrm{SFCL}}{}^{*}//R_{2} \end{array}$$

### 2.3. Current-Limiting Effect Analysis

Assuming that a 500 A/5 kA FC-SFCL works in the 10 kV single-phase system shown in Figure 2, the self-inductances *L*_1_ and *L*_2_ both are 7.5 mH, and the mutual inductance *M* is −7.499 mH. Here, the fault situation is set: the ground short circuit fault occurs at the first end of the transmission line, the fault resistance *R*_ground_ is 1.2 Ω, and the auxiliary circuit breaker operates within 20 ms. Table 1 shows the parameters of the simulation model of the system. The current values of the system under different quench resistances with the FC-SFCL are compared, and its current-limiting effect is analyzed. The quench resistance is simplified with a certain value resistance. Figure 3 shows the total current of the system under various working conditions, and the quench resistance is 0.01 Ω, 0.5 Ω, and 1 Ω, respectively. Table 2 shows the current value and current-limiting rate corresponding to the different working conditions.

The limiter has good current-limiting capability, and the current-limiting rate is above 50% in this model. For the non-quenched limiter, the fault current is limited by the current-limiting inductance after decoupling the parallel inductor. For the quenched limiter, the quench resistance before the decoupling of the parallel inductance can limit the first peak value of the fault current to a certain extent. The larger the quench resistance, the higher the current-limiting effect. After decoupling, the quench resistance and the current-limiting inductance of the parallel inductance jointly limit the fault current. Therefore, the quench-type parallel inductance can improve the overall current-limiting capability.

## 3. Research on the Simplified Calculation Method of the Number of Parallel Tapes

Before decoupling, the parallel inductance of the quenched FC-SFCL is similar to a non-inductive coil; after decoupling, the parallel inductance works as a current-limiting inductance. When designing the quenched limiter, the determination of the number of parallel tapes used in the parallel inductance is a key step. This section proposes a way to simplify the calculation of the number of parallel tapes used in the parallel inductance. The focus of the method is to verify whether the superconducting properties of superconductors can be ignored in the design process, which is studied from the perspective of tapes and coils.

### 3.1. Based Models

#### 3.1.1. Electromagnetic Field Equation Model

H-formulation is used to solve the magnetic field, and the two-dimensional Maxwell equations are used as the governing equations. The governing equations are derived from Maxwell’s equations for cylindrical coordinates [15,16] as
(3)$$\begin{array}{l} \left\{ \begin{array}{l} \mu \frac{\partial H_{r}}{\partial t}-\frac{\partial E_{\phi }}{\partial z}=0\\ \mu \frac{\partial H_{z}}{\partial t}+\frac{1}{r}\frac{\partial \left(rE_{\phi }\right)}{\partial r}=0 \end{array}\right.\\ J=\frac{\partial H_{z}}{\partial r}-\frac{\partial H_{r}}{\partial z} \end{array}$$

#### 3.1.2. Heat Transfer Model

The heat transfer model adopts the law of heat conduction [17], and its form is shown as:(4)$$\rho C_{p}\frac{\partial T}{\partial t}+\nabla \cdot \left(-k\nabla T\right)+\rho C_{p}u\cdot \nabla T=Q$$
where *ρ* is the density of the material; *C*_p_ and *k* are the specific heat capacity and thermal conductivity of the material, which are related to the physical properties of the material, and are quantities that change with time, temperature, or other parameters; *u* is the external field dependent variable, and the common quantities are speed, etc.; *T* is the Kelvin temperature; *Q* is the source term and represents the heat flux density. The heat source of superconducting the magnet mainly comes from the AC loss of the magnet, the eddy current loss of the metal structure, and the heat leakage of the current lead and Dewar. In this model, ignoring the influence of the external field, it is considered that the heat source mainly comes from the AC loss of the superconductor and the resistance heat of each part. 

#### 3.1.3. Electromagnetic-Thermal Coupling Model

Magnetic field and temperature both have an impact on tape’s critical characteristics. Considering them together, a Kim-like model [18] that can be used to describe the anisotropic tape, as shown in (5). *B*_⊥_ and *B*_//_ represent the magnetic field perpendicular and parallel to the tape direction, respectively. *A*, *b*, and *B*_0_ are the characteristic parameters of the tape, which can be obtained by fitting the data of the critical current from the experiment, generally 0 < *a* < 1. *J*_c_(*T*) is the critical current density of the tape at 77 K. *E*-*J* characteristics can be expressed by a power exponential function.
(5a)$$J_{c}\left(B,T\right)=\frac{J_{c}(T)}{{\left(1+\frac{\sqrt{a^{2}{\left|B_{//}\right|}^{2}+{\left|B_{⊥}\right|}^{2}}}{B_{0}}\right)}^{b}}$$
(5b)$$J_{c}(T)=J_{c}(77K)(\frac{T_{c}-T}{T_{c}-77K}{)}^{1.5}$$

In addition, the setting of boundary conditions is also very important for heat transfer. In the simulation, the boundary conditions that are most appropriate to the actual situation should be selected so that the results obtained are relatively accurate. Since this simulation is about immersion cooling, the boundary conditions are given in the form of heat flux curves.

Figure 4a shows the structure of a second-generation high-temperature superconducting tape. Here, a type of tape from Shanghai Superconductor Co. Ltd (Shanghai, China) was adopted. The size of the tape is 4 mm @ 0.21 mm, and the minimum critical current under self-field at 77 K is 100 A. Among them, the YBCO layer is 1 μm, the protective silver layer is 2 μm, the base Hastelloy layer is 50 μm, the single-sided reinforced copper layer is 75 μm, and the buffer layer is thin at 0.2 μm. A two-dimensional full model in Comsol Multiphysics [19] was built, as shown in Figure 4b, to simulate the current distribution and temperature rise characteristics of the tape.

During operation, the heat of the superconductor is the difference between the heat source and the heat dissipation. The heat source mainly comes from the hysteresis loss of the superconducting layer, the eddy current loss of the reinforcing layer, and the coupling loss between the layers. Heat dissipation includes heat taken away by the cooling medium, radiant heat, etc., with the former dominating. In order to clarify the influence of heat transfer on the temperature rise of superconductors, the calculation results of heat transfer and adiabatic models were compared.

The current-carrying capacity of superconductors is closely related to temperature and magnetic field. Essentially, temperature and magnetic field affect the flow capacity of the superconducting layer in the superconducting tape. However, this effect is only reflected during the quenching period of the superconductor. As the current, temperature, and magnetic field increase, part of the current in the superconducting layer transfers to other layers. After the superconductor completely quenches, the superconducting layer has almost no current-carrying capacity, and the current flows through other layers. The magnetic field no longer affects the current shunt of the superconductor. Before quenching, the current mainly flows through the superconducting layer, and the resistance is almost 0, and the loss is extremely low. After quenching, the quench resistivity of the superconducting layer is relatively high, and the current flows through other layers. It shows a certain resistance, and the loss increases. Under certain flow conditions, if the superconducting properties are not considered, that is the material properties of the superconducting layer are replaced by ordinary conductor properties, and its resistivity is a linear function *ρ*(*T*) that is only related to temperature. Considering the superconducting properties, the temperature rise of the superconductor is higher. In order to clarify the difference between the two conditions, the results of the model with and without superconductivity were compared.

### 3.2. The Overcurrent Model of Tapes

A 50 mm long tape was used, and the tape was immersed and cooled by 77 K liquid nitrogen. Four models for the tape were built: Model 1 (considering superconductivity and heat transfer), Model 2 (considering superconductivity and heat insulation), Model 3 (excluding superconductivity and heat transfer), and Model 4 (excluding superconductivity and heat insulation). The current-sharing and the temperature rise characteristics of the tape under the current amplitudes of 60 A, 80 A, 100 A, 200 A, 300 A, 400 A, 500 A, 600 A, and 700 A sinusoidal excitation were simulated. The heat transfer coefficient of the heat transfer boundary [20] is shown in Figure 5.

Figure 6 and Figure 7 are, respectively, the current diversion and the highest temperature value under 200 A and 600 A in Model 1 and Model 2. Model 1 considered the heat transfer power between the tape and liquid nitrogen, and Model 2 simulated adiabatic conditions. When the heat exchange was taken into account, the temperature rise was slightly lower. As the temperature increased, the current-carrying capacity of the superconducting layer YBCO decreased, the current was mainly transferred to the reinforced copper layer, and the current value gradually increased. When the temperature of the superconducting layer was higher than 92 K, the current in the YBCO layer was almost 0. The higher the tape flow was, the faster the quenching speed was.

Figure 8 shows the temperature rise of Models 1–4 under current amplitudes of 60 A, 80 A, 100 A, 200 A, 300 A, 400 A, 500 A, 600 A, and 700 A sinusoidal excitation. As the current increased, the tape’s temperature rose faster. Compared to the heat transfer boundary, the tape’s temperature rose faster with the adiabatic boundary. Compared to superconductivity, the tape’s temperature rose slightly faster without considering superconductivity.

Figure 9 shows the temperature rise difference on the tape under different current-carrying conditions between Model 1 and Model 2, and between Model 1 and Model 3. The legend “detT + value” indicates the temperature difference between the two calculation models under the current excitation of the value. The temperature rise difference under adiabatic and heat transfer boundary was larger. Under the conditions, considering superconductivity or not, the temperature rise difference was smaller, all of which were lower than 6 K. The temperature rise difference increased with the current value below 300 A and had a downward trend when the current was higher than 400 A. This is because, under low current, the superconducting layer has current-carrying capacity, along with lower heat. When the current is large, the speed of the superconducting layer losing current-carrying capacity becomes fast as the current increases, and the tape’s working state is closer to the state of ignoring superconductivity.

In summary, the overcurrent temperature rise data were valid when ignoring the superconductivity of the tape. The impact boundary curve of the tape, which was calculated according to the normally conducting wire while ignoring the superconductivity, was used to determine the overcurrent parameters of the tape and calculate the number of parallel tapes of the parallel inductance.

### 3.3. The Overcurrent Model of Coils

Based on the above analysis of the superconducting tape, the temperature of the tape rose faster under adiabatic conditions. Furthermore, when the superconductor was located under a certain magnetic field, the superconducting characteristics of the superconductor gradually attenuate with the increase of the magnetic field, and the attenuation speed becomes faster and faster, until the superconducting characteristic disappears. Here, Model A (considering superconductivity) and Model B (excluding superconductivity and taken as normal conducting wire) are defined. Thus, the temperature rise characteristics of the tape under the two conditions were closer. From this, it can be concluded that the temperature rise of coils under the two conditions is closer. Related examples are given below.

A small coil with 16 turns was taken as the analysis object. The coil is made of tape wrapped with a layer of 70 µm thick polyimide film. The inner radius of the coil is 82.5 mm and its model is shown in Figure 10. The coil’s inductance is 104 μH, and the critical current is 62 A.

Figure 11 shows the change curve of the highest temperature value on the coil with time under different current amplitudes. Figure 11a corresponds to that of Model A and (b) corresponds to that of Model B. The time required to reach the quenched temperature 92 K was shorter when superconductivity is not considered. Figure 12 extracts the maximum magnetic flux density value on the coil under the different current amplitudes, and this value was proportional to the current amplitude.

Compared to the tape, the temperature difference of the coil was smaller under Models A and B, which was lower than 2.5 K, as shown in Figure 13a. When the current was lower than 300 A, the temperature difference increased as the current value. When the current was higher than 400 A, the temperature difference had a downward trend. The change rule is similar to that of the tape. As the current on the coil increased, the self-field effect became stronger, and the magnetic field accelerated the attenuation of the superconducting properties of the superconducting coil. Therefore, the temperature rise of the coil under Models A and B was closer than that of the tape.

Compared to the tape, the temperature difference of the coil was smaller under Models A and B, which was lower than 2.5 K, as shown in Figure 13a. When the current was lower than 300 A, the temperature difference increased with the current value. When the current was higher than 400 A, the temperature difference had a downward trend. The change rule is similar to that of the tape. As the current on the coil increased, the self-field effect became stronger, and the magnetic field accelerated the attenuation of the superconducting properties of the superconducting coil. Therefore, the temperature rise of the coil under Models A and B was closer than that of the tape.

For superconducting coils, the greater the inductance value, the greater the magnetic flux density generated by the overcurrent; thus, the more severe the critical current attenuation, the closer the temperature rise is to the normally-conducting coil. Here, take the normally conducting coil as an example to analyze the relationship between temperature rise and coil size. Assuming that the inner diameter of the coils is the same, the larger the coil’s scale and the more corresponding turns, the greater the coil’s inductance and the amount of wire used. Ignoring the heat exchange between the coil and liquid nitrogen, the relational expression of temperature rise is derived, as shown in (6). It can be seen that the coil temperature rise is not directly related to the amount of wire used; thus, the coil temperature rise is not directly related to the size of the coil.
(6a)cmΔT=QQ=RI2,m=dV=dSl,R=ρlS
(6b)ΔT=ρI2cdS2

To verify whether the temperature rise of the normally conducting coil was related to the coil size, a 50-turn coil model was established and compared with the 16-turn coil. Among them, the inductance of the 50-turn coil was 856 μH. Figure 13b shows the temperature difference of the 16-turn coil and the 50-turn coil under different currents. It shows that the temperature difference between the two coils was extremely small, both within 0.1 K. There were some fluctuations in the initial stage and stable oscillations at or near a certain value in the later stage, which verified the theoretical derivation.

In summary, the overcurrent characteristics of tapes and coils, whether considering superconducting characteristics or not, were analyzed from the perspective of temperature rise. For the tape, the temperature difference was less than 6 K under the two calculation conditions; when the current was higher than 300 A, the temperature rise under the two conditions was closer. For the coil, the temperature difference was lower under the two conditions because the self-field effect of the coil was stronger than that of the tape, and the magnetic field accelerated the attenuation of the superconducting characteristics of the superconducting coil. As the coil size increased, the temperature gap became smaller. Based on the normal conducting wire, the impact boundary of the tape was calculated, the overcurrent parameters were selected according to the actual demand, and then the number of parallel tapes of the parallel inductance was determined. If it is estimated according to adiabatic conditions, the design margin reserved is greater. The above work verifies the feasibility of the simplified calculation of the number of parallel tapes.

## 4. The Electromagnetic Design Method of the Quenched FC-SFCL

The quenched FC-SFCL means that the limiter quenches when the system fault current is large. Therefore, the current-carrying capacity of the limiter should be based on the rated current and its ability to withstand the maximum current.

Aiming at the design of the quench-type parallel inductance, this paper proposed a design method based on the simplified calculation of the number of parallel tapes. The key point was to calculate the number of parallel tapes according to the simplified calculation method, i.e., to ignore the superconducting characteristics and calculate the impact boundary curve of the tape according to the normal conducting wire. First, the allowable over-current multiples were selected according to the over-current requirements. Then, the number of parallel tapes required for the parallel inductance was determined according to the maximum through-current and allowable over-current multiples. Last, the electromagnetic optimization design was carried out according to conventional methods. The previous design idea was to obtain the number of parallel tapes in the electromagnetic optimization process, and the magnetic field anisotropy of the tape needs to be considered. This method obtain the number of parallel tapes first and then optimized the design. The influence of the magnetic field can be ignored and the design difficulty can be reduced. The overall design process is shown in Figure 14.

## 5. The Electromagnetic Design Example of the Quenched FC-SFCL 

### 5.1. The Design Example

A 10 kV/500 A quenched limiter prototype was designed by the electromagnetic optimization design method in Chapter 4. The non-quenched prototype with the same parameters was designed in [12]. The expected peak fault current was 5 × sqrt(2) kA, and the fault steady-state current after current limiting was 2.5 kA.

Here, we used the Superpower SCS4050 tape. The reinforced layer is the key factor to influence the overcurrent-carrying characteristics of the tape. The simulation about overcurrent endurance time was carried out on the tapes with different reinforced layer thicknesses, and the ultimate endurance time under different maximum allowable temperatures was obtained, which is shown in Figure 15. Among them, the heat exchange between the tape and the liquid nitrogen was ignored, and the self-field critical current 122 A at 77 K was used as the standard value. The calculated result was used as the selection criterion for the overcurrent multiple, reversing enough of the design margin. In the design, the maximum allowable temperature was 300 K. According to the operation speed of the switch, the limit withstand time of the tape was required to be 120 ms. Thus, the 100 μm thickness reinforced layer of the tape was selected, which allowed for a five-times higher overcurrent [21].

The operating temperature zone of the limiter was selected as 77 K. The number of parallel tapes required for the parallel inductance was calculated according to the expected current peak value and the fault steady-state current after current limiting, and the larger value was taken. Then, it was checked whether the current carrying capacity of the parallel inductance exceeded the allowable current margin under rated conditions. If not, the design requirements were met, otherwise the number of parallel tapes would be increased until the design requirements were met.

According to the expected current peak value, the total number of parallel tapes of the parallel inductance was 5 × sqrt(2) × 1000/122/5 ≈ 12, and the number of parallel tapes of a single branch in the parallel inductance was 2.5 × sqrt(2) × 1000/122/5 ≈ 6, according to the fault steady-state current after current limiting. Tentatively, the parallel number of the two branches of the parallel inductance was 6, and the allowable current margin was 0.8; thus, 6 × 2 × 122 × 0.8 = 1170 A > 500 × sqrt(2) A. Therefore, the above design met the requirements.

Based on the optimized electromagnetic parameters in Section 4, the number of parallel tapes of superconducting coils was modified, it was verified whether the working state of the parallel inductance under rated conditions met the requirements through a specific model, and its loss value and temperature rise were evaluated.

First, the highest magnetic flux intensity on the superconducting coil at rated state was calculated, which was only 0.037 T; at this time, the minimum current-carrying capacity of a single tape was 110 A. Therefore, the total current-carrying capacity of the superconducting parallel inductance was not less than 110 × 6 × 2 = 1320 A, which is higher than the rated value and met the requirements of safe operation.

Based on the H-formulation method, the AC loss calculation model of the quenched superconducting parallel inductance at the rated state was built. Figure 16 shows the total AC loss and the two branch losses of the parallel inductance at the rated state. At this stage, the peak loss was less than 0.5 W, and the cooling power easily met the demand.

In order to evaluate the temperature rise of the parallel inductance during faults, an electromagnetic-thermal-coupling model was built based on normally conducting coils instead of superconducting coils for a rough check. Coupling the magnetic field (mf) module and the solid heat transfer (ht) module in Comsol Multiphysics, the calculation model of the parallel inductance was established, and the temperature rise during the fault under the adiabatic boundary was calculated. The current waveforms of the two branches are shown in Figure 5 in [12], and the fault after decoupling of the parallel inductance lasts for one more cycle. Figure 17 shows the relationship curve between currents and maximum temperatures of the parallel inductance during a fault. Under this working condition, the maximum temperature of parallel inductance is still less than 115 K. Therefore, the above design scheme is feasible.

### 5.2. Comparison of Quenched and Non-Quenched Schemes

In the comparison case, each branch of the non-quenched current limiter had 18 tapes in parallel (65 K), and the quenched one only needed 6 tapes (77 K); thus, the wire consumption was greatly reduced. The quenched scheme had outstanding technical and economic efficiency. In terms of steady-state operating loss, the number of parallel tapes required for the parallel inductance of the quenched current limiter was reduced, the total current carrying capacity of the coil was reduced, and the allowable current carrying margin was reduced. The corresponding steady-state AC loss was larger than the non-quench type. In terms of system operation stability, the non-quench current limiter did not quench under any working conditions, and there was no quench recovery problem. It can cooperate with the automatic reclosing of the power system, and the system stability was higher. During the system fault, the quenched limiter has different degrees of quench. After the system fault is cut off, it should be ensured that the limiter restores to the superconducting state before reconnecting to the system. Table 3 summarizes the advantages and disadvantages and applicable scenarios of the two working schemes of the current limiter.

The electromagnetic design of the non-quench type and quench type FC-SFCL was discussed and exemplified in detail above. The two working schemes of the limiter had many similarities in the electromagnetic design process, and the biggest difference was in the determination of the number of tapes in parallel. The non-quench type needs to ensure that there is no quench phenomenon during operation, and its current-carrying capacity under the maximum leakage magnetic field still had a certain margin compared to the maximum working current. In the electromagnetic design, the critical current-carrying capacity under the maximum fault current at the fault state and current-limiting state was comprehensively considered. This value is closely related to the magnetic flux intensity under the corresponding working conditions. It is necessary to ensure that the value is higher than the working current to determine the number of tapes in parallel. The focus of the electromagnetic design of the quench type limiter is to ensure the rated operation of the parallel inductance without quench and to safely and stably operate under fault conditions. The current-carrying capacity of the limiter should be based on the design requirement that no overcurrent occurs at the rated current and the maximum allowable current that it can withstand to determine the number of tapes in parallel. The work schemes of the limiter should be selected according to the actual application scenario requirements.

## 6. Conclusions

In this paper, the engineering implementation scheme of the FC-SFCL was explored and studied. In order to improve the technical economy of the engineering prototype, the quench-type improvement scheme was proposed, and the research work was carried out for its electromagnetic design method. The specific works and conclusions are as follows:(1)Considering that the quench-type parallel inductance can limit the first peak value of the fault current, a quench-type improvement scheme was proposed. The scheme can limit the first peak value of fault current to a certain extent by the quench resistance at the initial stage of the fault. After the parallel inductance is decoupled, the quench resistance and the current-limiting inductance jointly limit the fault current, which improves the current limiting capability of the limiter and reduces the interruption requirements of the circuit breakers.(2)In order to reduce the design difficulty of the quenched FC-SFCL, a simplified calculation idea of the number of parallel tapes and a design method based on the simplified calculation idea of the parallel inductance were proposed. The idea is to ignore the superconducting characteristics and calculate the impact boundary curve of the tape according to the normal conducting wire. Then the allowable over-current multiples were selected according to the over-current requirements, and the number of parallel tapes of the parallel inductance were determined according to the maximum through-current and allowable over-current multiples. This design method is based on the idea of simplifying the calculation of the number of parallel tapes, which can ignore the influence of magnetic field, simplify the design process, and reduce the design difficulty.(3)Taking the 10 kV/500 A/5 kA prototype as an example, the electromagnetic design of quenched parallel inductance was completed, and the performance of the two schemes was compared. Compared to the non-quenched structure, the technical economics of the quenched one were more prominent, and it can be used preferentially for engineering prototypes.

## Figures and Tables

**Figure 1 materials-16-00754-f001:**
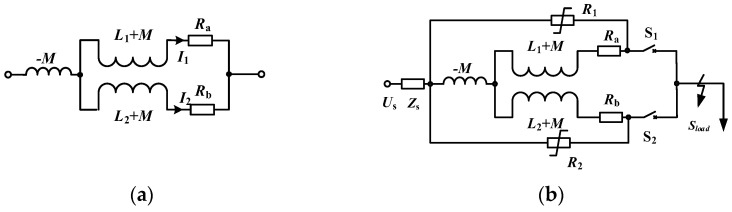
The equivalent circuit of the quenched FC-SFCL. (**a**) The equivalent circuit of the parallel inductance; (**b**) equivalent circuit of the FC-SFCL.

**Figure 2 materials-16-00754-f002:**
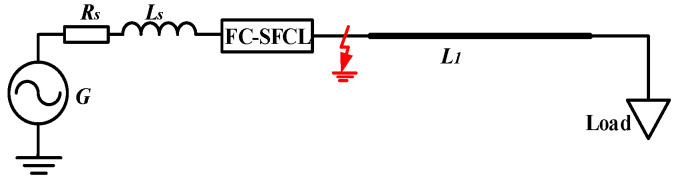
The 10 kV single-phase system.

**Figure 3 materials-16-00754-f003:**
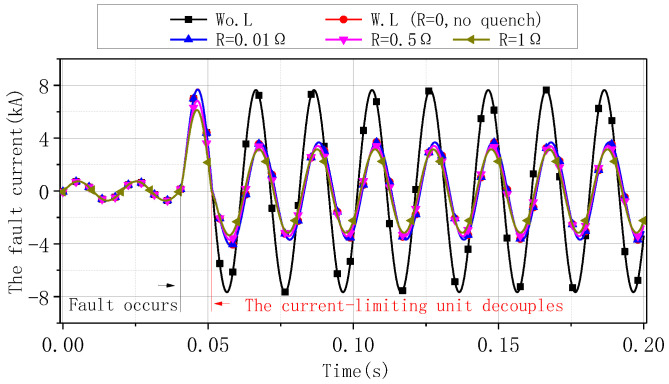
Current waveforms under different conditions.

**Figure 4 materials-16-00754-f004:**
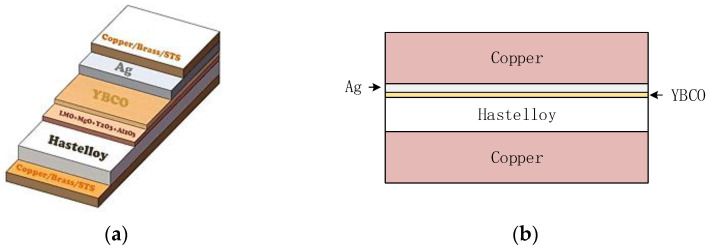
The structure of superconducting tape. (**a**) The structure of the 2G HTS tape; (**b**) 2D full model in Comsol Multiphysics. Here, the buffer layer is omitted because its thin width and poor conductivity affect the characteristics of the tape.

**Figure 5 materials-16-00754-f005:**
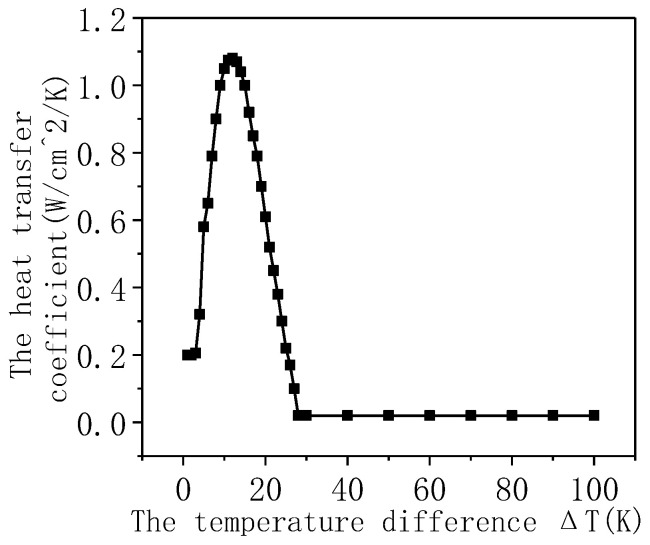
The heat transfer coefficient of LN2.

**Figure 6 materials-16-00754-f006:**
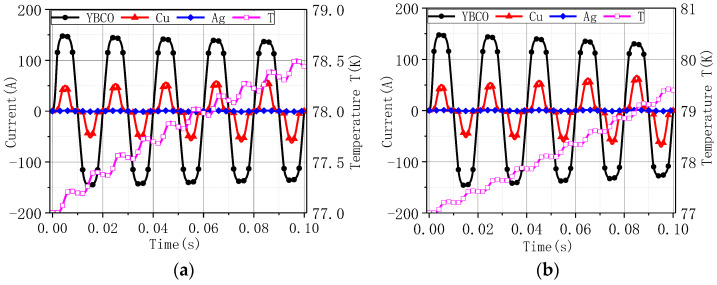
The current diversion and the highest temperature value under 200 A: (**a**) Model 1; (**b**) Model 2.

**Figure 7 materials-16-00754-f007:**
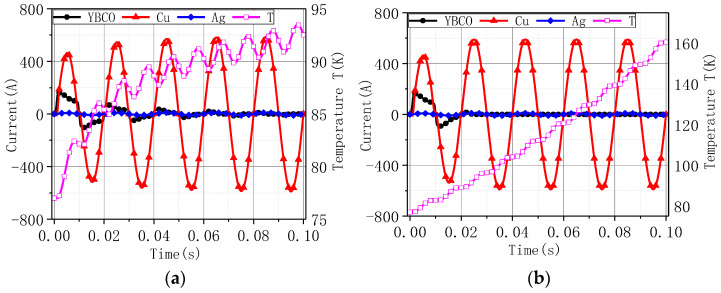
The current diversion and the highest temperature value under 600 A: (**a**) Model 1; (**b**) Model 2.

**Figure 8 materials-16-00754-f008:**
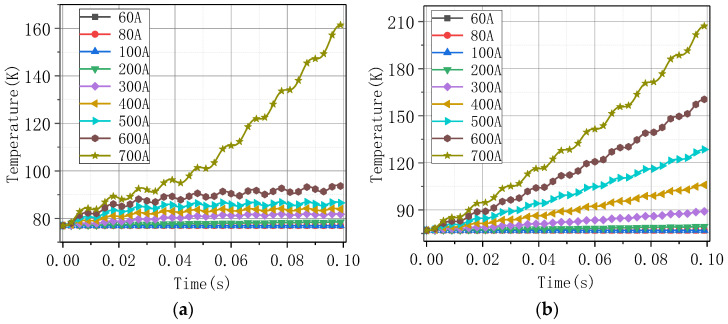
The highest temperature value under different current amplitudes: (**a**) Model 1; (**b**) Model 2; (**c**) Model 3; (**d**) Model 4.

**Figure 9 materials-16-00754-f009:**
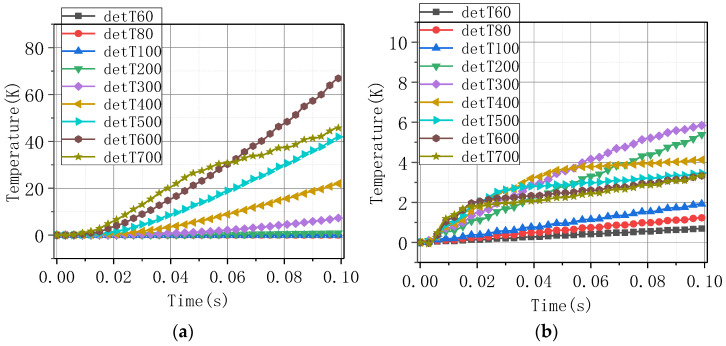
The highest temperature difference under different current amplitudes: (**a**) Model 2–Model 1; (**b**) Model 3–Model 1.

**Figure 10 materials-16-00754-f010:**
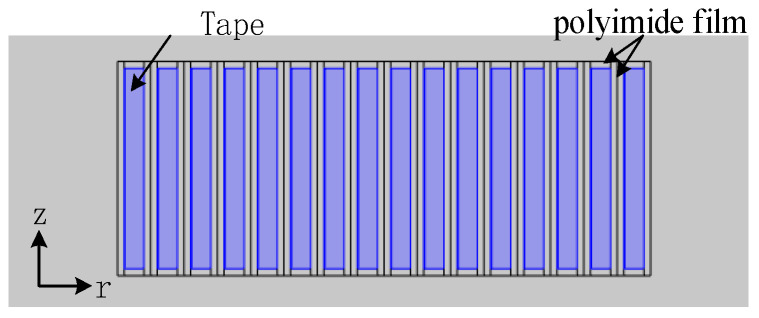
The model of the 16-turn coil.

**Figure 11 materials-16-00754-f011:**
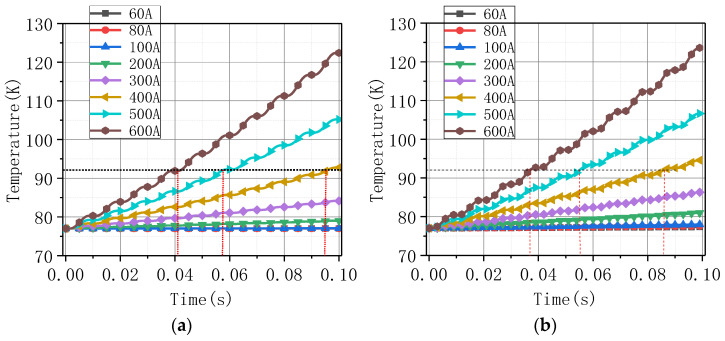
The highest temperature under different current amplitudes: (**a**) Model A; (**b**) Model B. The black dashed line represents the quenched temperature 92 K, and the red dashed lines represent the time used to reach the quenched temperature.

**Figure 12 materials-16-00754-f012:**
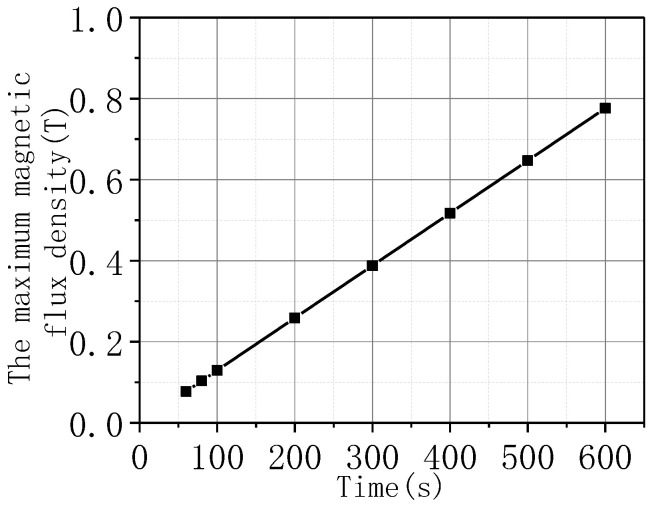
The maximum magnetic flux density value under different current amplitudes.

**Figure 13 materials-16-00754-f013:**
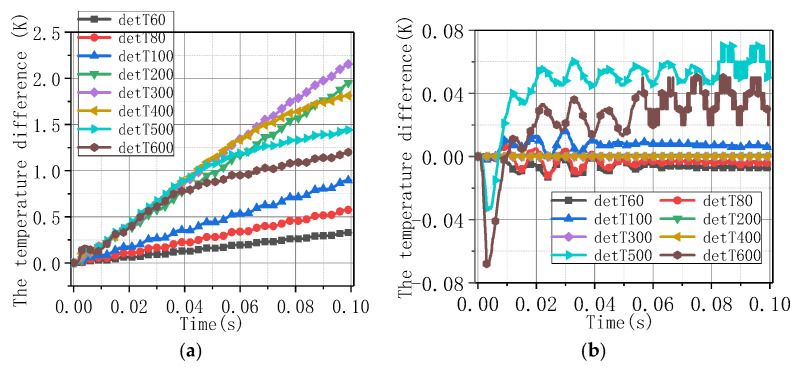
The highest temperature difference under different current amplitudes: (**a**) under Model A and B; (**b**) the 16-turn coil–the 50-turn coil.

**Figure 14 materials-16-00754-f014:**
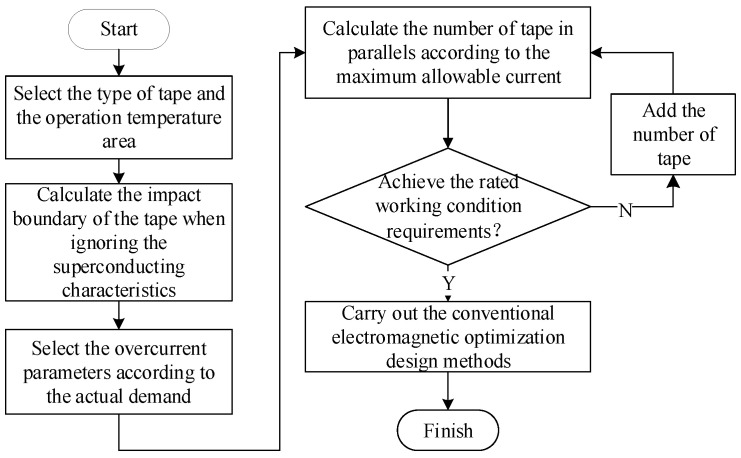
The electromagnetic optimization design method of the quenched limiter.

**Figure 15 materials-16-00754-f015:**
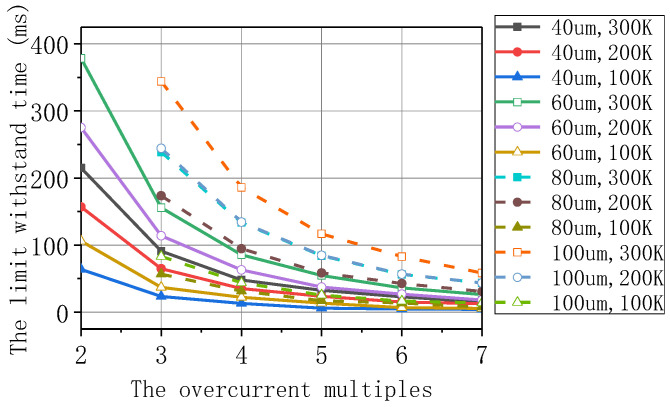
Impact boundary curves under different reinforcement layer thicknesses.

**Figure 16 materials-16-00754-f016:**
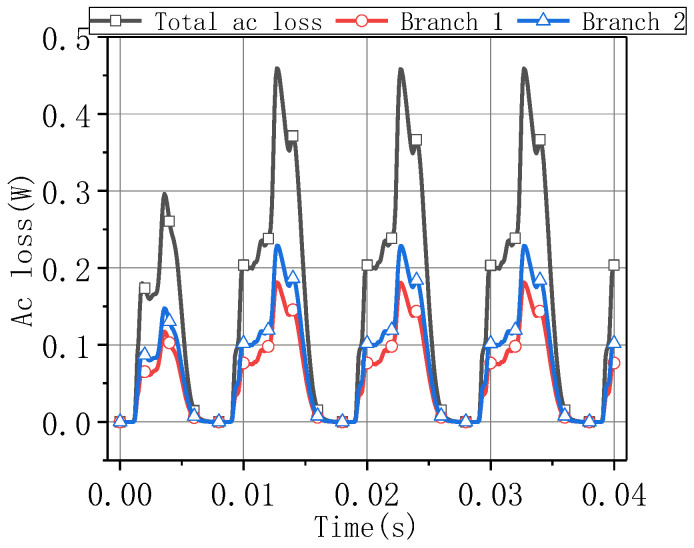
AC loss during rated operation.

**Figure 17 materials-16-00754-f017:**
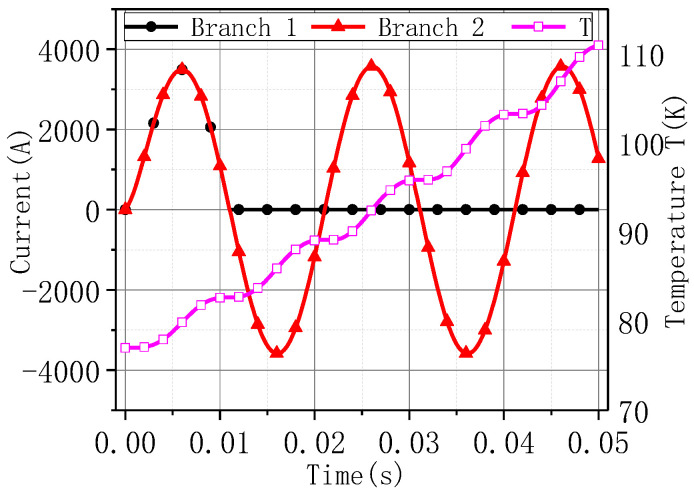
Current and temperature curves of parallel inductance during a fault.

**Table 1 materials-16-00754-t001:** Parameters of the simulation model.

Terms	Parameters
Generator	*V*_n_ = 10 kV, *f* = 50 Hz
Transmission line	*R*_l_ = 0.512 Ω, *L*_l_ = 2.384 mH, *C*_l_ = 86.6 nF
Load	*V*_n_ = 10 kV, *P*_n_ = 5.0 MW, *Q*_n_ = 280 kVar

**Table 2 materials-16-00754-t002:** Current value and current-limiting rate under different conditions.

Conditions	Wo.L	W.L (the Value of *R*)			
		0	0.01 Ω	0.5 Ω	1 Ω
*I*_max_ in state 2 (kA)	7.70	7.70	7.68	6.83	6.13
*I*_max_ in state 3 (kA)	7.66	4.14	4.13	4.05	3.35
*I*_rms_ in state 3 (kA)	5.41	2.61	2.60	2.41	2.23
*δ*% 1st peak current	/	0.0%	0.3%	11.3%	20.3%
*δ*%_max_	/	45.9%	46.0%	47.1%	56.3%
*δ*%_rms_	/	51.8%	51.9%	55.4%	58.7%

Note: (1) State 2 and 3 are, respectively, the fault condition and the current-limiting condition. (2) Wo.L represents the power system without a limiter, and W.L represents the power system with a limiter.

**Table 3 materials-16-00754-t003:** Comparison of advantages and disadvantages of non-quench type and quench type SFCL.

Terms	Non-Quench Type	Quench Type
Advantages	(1) No quench recovery problem; (2) Cooperate with system automatic reclosing; (3) Large current margin and low loss during rated operation.	(1) Saves tape and reduces cost; (2) Quench resistance can limit the peak value of fault current.
Disadvantages	(1) Uses a large number of tapes; (2) Cost is high; (3) Cannot limit the peak value of the fault current.	(1) Has a certain degree of quench recovery problem; (2) Cannot effectively cooperate with system automatic reclosing; (3) Higher refrigeration requirements.
Applicable scenarios	Which can effectively remove faults, improve system stability and recovery automatically without limiting the first peak value of fault current	Which needs the limiter respond automatically and limit the first peak value of the fault current and has high technical and economic efficiency

## Data Availability

The data that support the findings in this study are available from the corresponding author upon reasonable request.

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
