# Peer review of "Quenched Flux-Coupling Superconducting Fault Current Limiter Scheme and Its Electromagnetic Design Method"

_materials, 2023, doi:10.3390/ma16020754_

Round 1
Reviewer 1 Report
referee report
materials-2122246-peer-review-v1
Quenched flux-coupling superconducting fault current limiter scheme and its electromagnetic design method
Sinian Yan et al.
The present manuscript discusses the properties of a flux-coupling type superconducting fault current limiter and
its electromagnetic design by simulations using COMSOL. This topic is interesting in the view of the given Special
Issue of Materials.
The present manuscript comprises 17 figures, 3 tables and 20 references are provided. The text is well written (also
in quite proper English) and all figures included are well prepared.
The authors provide a proper discussion of the various types of FCLs in the introduction, and discuss the operating
principle of the FC-SFCL and its equivalent circuit. Then, different models for analysis are presented.
There are several points which require attention:
--Please take care for spaces in the entire manuscript. There should always be a space between a physical quantity and
its unit, but also between text and references, text and included brackets, etc.
--Physical quantities in the text should be written always in italics (like in the formulae); subscripts, which are
not just numbers, in roman. This also applies for constants like k and n below eq.(1) or line 153
-- Sec. 3.1: "H-formulation" -- H is here a vector, and ths, should be written in bold.
--line 163: adopted
--Fig. 13: Please explain the meaning of the red and black dashed lines also in the figure caption.
--Please reformat Table 3. If the space is not sufficient, please introduce a long table. In the present form, it
is quite difficult to understand which entry belongs to what.
--Ref.2: Article no. 8754
--Ref.4: page/article no. is missing
--Refs. 14, 15: page/article no. is missing
Overall, the present manuscript contains useful information to promote the SC-SFLC current limiter based on YBCO tapes, and fits well to the scope of the Special Issue. Thus, the manuscript is suitable for publication with minor revisions.
Author Response
Dear Reviewer,
RE: Manuscript materials-2122246 titled “Quenched flux-coupling superconducting fault current limiter scheme and its electromagnetic design method”.
We thank you for their careful read and thoughtful comments on the previous draft. We have carefully taken your comments into consideration in preparing our revision, which has resulted in a paper that is clearer, more compelling, and broader. The revisions made to previous draft are marked up using the “Track Changes” function of MS Word. The following summarizes how we responded to your comments.
Thanks for all the help.
Best wishes,
Dr. Sinian Yan (Corresponding Author: professor Jinghong Zhao)
Revision — authors’ response
We have studied your comments carefully and improved the manuscript. We appreciate for your warm work earnestly. The revisions of this paper are summarized as follows and the modified places were marked up using the “Track Changes” function of MS Word in the revised draft.
Comment: Please take care for spaces in the entire manuscript. There should always be a space between a physical quantity and its unit, but also between text and references, text and included brackets, etc.
Response: Thanks for your kind comments. We have amended them as you pointed out. We have supplemented spaces at the lost locations.
Comment: Physical quantities in the text should be written always in italics (like in the formulae); subscripts, which are not just numbers, in roman. This also applies for constants like k and n below eq.(1) or line 153.
Response: Thanks for your kind comments. We have amended them as you pointed out.
Comment: Sec. 3.1: "H-formulation" -- H is here a vector, and ths, should be written in bold.
Response: We have amended them as you pointed out.
Comment: line 163: adopted
Response: We have amended them as you pointed out.
Comment: Fig. 13: Please explain the meaning of the red and black dashed lines also in the figure caption.
Response: Thanks for your constructive suggestion. We have supplemented the meaning of the red and black dashed lines in the caption of Figures 11 and 13. Now the draft is much clearer after modification.
Comment: Please reformat Table 3. If the space is not sufficient, please introduce a long table. In the present form, it is quite difficult to understand which entry belongs to what.
Response: Thanks for your constructive suggestion. In the original draft, it is really confusing. We have modified Table 3 according to your suggestions.
Comment: Ref.2: Article no. 8754
Ref.4: page/article no. is missing
Refs. 14, 15: page/article no. is missing
Response: We have amended them as you pointed out.
In addition to the comments you proposed have been modified, we also revised the formats of several texts in previous version, which have been marked up using the “Track Changes” function of MS Word in the revised draft.
It is a great honor for us get the your good comments on our manuscript. Please feel free to contact us with any questions and we are looking forward to your consideration.
Thanks for your consideration again.
Sincerely,
Sinian Yan

Reviewer 2 Report
Fault current limiters are the perspective niche for superconducting applications. Various types of superconducting fault current limiters have a different balance of efficiency and economics and make different impacts on a circuit in the rated and fault states. Authors compared quenched and non-quenched schemes of flux-coupling superconducting fault current limiter. Efficiency and economy of consumed superconducting tapes were justified for the quenched scheme. The original simplified calculation method was suggested and applied to design the quenched flux-coupling SFCL. To reduce the design difficulty, just normal properties of superconducting tapes were accounted for the number of parallel tapes. The paper is well written. The text is clear and easy to read. The conclusions consistent with the evidence and arguments. The manuscript may be published.
Author Response
Dear Reviewer,
RE: Manuscript materials-2122246 titled “Quenched flux-coupling superconducting fault current limiter scheme and its electromagnetic design method”.
Thanks for your positive comments and appreciation on our draft. We would like to appreciate your recognition again.
The latest draft has made some revisions according to the comments from other reviewers. We wish the draft is clearer, more compelling, and broader. The revisions made to previous draft are marked up using the “Track Changes” function of MS Word.
It is a great honor for us get the your review on our manuscript. Please feel free to contact us with any questions and we are looking forward to your consideration.
Thanks for all the help.
Best wishes,
Dr. Sinian Yan (Corresponding Author: professor Jinghong Zhao)
Reviewer 3 Report
1. What is the circuit diagram that is described by equation (1)?
2. Why is the coupling coefficient in Figure 1 denoted by M and in the text and formula (1) by k?
3. In Table 2, the current symbol is written in italic font. The max and rms symbols should be subscripts.
4. On line 153, there should be a B0 symbol with 0 in the subscript.
5. The superconducting tape structure shown in Figure 4a is poorly visible. I suggest drawing the drawing from the beginning or downloading the tape construction drawing from the manufacturer's website.
6. Why is the buffer layer between the superconductor layer and the hastelloy layer omitted from the model used for simulation in Comsol, shown in Figure 4b? The manufacturer on the website states the construction of this layer. Does the buffer layer not affect the temperature distribution in the superconducting tape?
Author Response
Dear Reviewer,
RE: Manuscript materials-2122246 titled “Quenched flux-coupling superconducting fault current limiter scheme and its electromagnetic design method”.
We thank you for their careful read and thoughtful comments on the previous draft. We have carefully taken your comments into consideration in preparing our revision, which has resulted in a paper that is clearer, more compelling, and broader. The revisions made to previous draft are marked up using the “Track Changes” function of MS Word. The following summarizes how we responded to your comments.
Thanks for all the help.
Best wishes,
Dr. Sinian Yan (Corresponding Author: professor Jinghong Zhao)
Revision — authors’ response
We have studied your comments carefully and improved the manuscript. We appreciate for your warm work earnestly. The revisions of this paper are summarized as follows and the modified places were marked up using the “Track Changes” function of MS Word in the revised draft.
Comment 1: What is the circuit diagram that is described by equation (1)?
Response: Thanks for your kind comments. We were sorry for the unclear description about the circuit diagram and equation (1). In equation (1), L1 and L2 are the self-inductance of the two superconducting coils, M is the mutual inductance, k=M/√(L1L2) and n=√(L1/L2) are the coupling coefficient and the transformation ratio of the two coils, respectively, which corresponds to the circuit diagram shown as Figure 1a. We have supplemented the related description in the first and second paragraphs in section 2.2.
Comment 2: Why is the coupling coefficient in Figure 1 denoted by M and in the text and formula (1) by k?
Response: Thanks for your good comments. The original draft lacks the description of the relationship between M and k. M is the mutual inductance of the two superconducting coils and k the coupling coefficient of the two coils. There has k=M/√(L1L2), where L1 and L2 are the self-inductance of the two superconducting coils. We have supplemented the related description in the first and second paragraphs in section 2.2 and the text below formula (1).
Comment 3: In Table 2, the current symbol is written in italic font. The max and rms symbols should be subscripts.
Response: Thanks for your kind comments. We have amended them as you pointed out.
Comment 4: On line 153, there should be a B0 symbol with 0 in the subscript.
Response: We have amended them as you pointed out.
Comment 5: The superconducting tape structure shown in Figure 4a is poorly visible. I suggest drawing the drawing from the beginning or downloading the tape construction drawing from the manufacturer's website.
Response: Thanks for your kind comments. We have amended the superconducting tape structure shown in Figure 4a to be clearer.
Comment 6: Why is the buffer layer between the superconductor layer and the hastelloy layer omitted from the model used for simulation in Comsol, shown in Figure 4b? The manufacturer on the website states the construction of this layer. Does the buffer layer not affect the temperature distribution in the superconducting tape?
Response: Thanks for your good question. In the original draft, the question is really not described. The buffer layer between the superconductor layer and the hastelloy layer is about 0.2 μm. Because of its thin width and poor conductivity, the effect to the characteristics of the tape can be omitted. So the buffer layer is omitted from the model used for simulation in Comsol, shown in Figure 4b. We have supplemented the reason in the caption of Figure 4.
In addition to the comments you proposed have been modified, we also revised the formats of several texts in previous version, which have been marked up using the “Track Changes” function of MS Word in the revised draft.
It is a great honor for us get the your good comments on our manuscript. Please feel free to contact us with any questions and we are looking forward to your consideration.
Thanks for your consideration again.
Sincerely,
Sinian Yan
